

# Impact of boundary layer stability on urban park
# cooling effect intensity
*Authors: Martial Haeffelin[1], Jean-François Ribaud[2], Jonnathan Céspedes[2], Jean-*
*Charles Dupont[3], Aude Lemonsu[4], Valéry Masson[4], Tim Nagel[4], Simone Kotthaus[2]*
[1] *Institut Pierre Simon Laplace (IPSL), CNRS, Ecole polytechnique, Institut Polytechnique de Paris, 91128*
*Palaiseau Cedex, France*
[2] *Laboratoire de Météorologie Dynamique (LMD-IPSL), Ecole polytechnique, Institut Polytechnique de*
*Paris, 91128 Palaiseau Cedex, France*
[3] *Institut Pierre Simon Laplace (IPSL), Université Versailles Saint-Quentin-en-Yvelines, 78240*
*Guyancourt, France*
[4] *Centre national de recherches météorologiques (CNRM), Université de Toulouse, Météo-France,*
*CNRS, Toulouse, France*
Correspondence to: Martial Haeffelin (martial.haeffelin@ipsl.fr)





# Abstract

The added heat in cities amplifies the health risks of heat waves. At night under calm winds and cloud free skies, the air in the urban canopy layer can be several degrees warmer than in rural areas. This lower nocturnal cooling in the built-up settings poses severe health risks to the urban inhabitants as indoor spaces cannot be ventilated effectively. With heat waves becoming more frequent and more intense in future climates, many cities are expanding their green spaces with the aim to introduce cooling through shading, evaporation, and lower heat storage capacities. In this study, it is assessed how the evening and night-time cooling effect of urban parks (relative to near-by built-up settings) varies with the park size and the meso-scale atmospheric conditions during warm summer periods. Using a combination of meteorological surface station data and compact radiosondes, the cooling effect is quantified for several urban parks (about 15 ha) and urban woods (about 900 ha). A profiling Doppler wind lidar deployed in the city centre is used to measure turbulent vertical mixing conditions in the urban boundary layer. We find that the maximum nocturnal cooling effects in urban parks range around 1-5°C during a one-week heat wave event in mid-July 2022 but also in general during summer 2022 (June-August). Three atmospheric stability and mixing regimes are identified that explain the night-to-night variability in park cooling effect. We find that very low turbulent vertical mixing in the urban boundary layer (< 0.05 $m^2s^{-2}$) results in the strongest evening cooling in both rural settings and urban parks and the weakest cooling in the built-up environment. This regime specifically occurs during heat waves in connection with large-scale advection of hot air over the region and corresponding subsidence. When nocturnal turbulent vertical mixing above the city is stronger, the evening cooling in urban green spaces is less efficient so that the atmospheric stratification above both urban parks and woods is less stable and temperature contrasts compared to the built-up environment are less pronounced. These results highlight that urban green spaces have a significant cooling potential during heat waves, with maximum effects at night as advection and mixing transport processes are minimal. This suggests adapting the opening hours of public parks to enable residents to benefit from these cooling islands.



## 1 - Introduction

Excess heat in cities has impacts on human comfort, labour productivity, and health. Mortality has been linked to exceptionally high temperatures during summertime heat waves both at night and during the day (Basu et al. 2002; Keatinge et al. 2000; Pirard et al. 2005). During the day, it is the outdoor radiative temperature that poses the most significant health risk. At night, indoor temperatures are particularly important as people need to rest and indoor air must be vented to cool the building for the upcoming day. However, urban inhabitants can be particularly exposed to excessive and prolonged heat stress at night as the city and the buildings do not cool efficiently, preventing necessary nocturnal rest. Hot nights following hot days have been shown to make an important contribution to heat-related mortality (Murage et al. 2017; Royé et al., 2021).

Reducing people's exposure to heat in cities can be addressed through urban planning strategies. Increasing the vegetation fraction of urban areas is a widely accepted strategy to mitigate urban heat risk by effectively reducing heat storage uptake during daytime (Grimmond and Oke, 2002). Trees can provide efficient shading whereby reducing daytime air temperatures by several degrees below their canopy, while evapotranspirative cooling provided by vegetation, including trees, shrubs and grass, maintain the green space temperature several degrees below that of the built-up environment (Shashua-Bar and Hoffman, 2000). The green infrastructures also show cooling effects at night, through continued evapotranspiration after sunset, generally larger sky-view factors in urban parks than in built-up environments, and lower heat capacities. However, reduced radiative cooling and ventilation can retain heat below the canopy at night (Taha et al. 1991).

The cooling effect intensity of urban green infrastructure has been shown to be highly variable (Bowler et al. 2010; Shoulika et al 2014). Doick et al (2014) point to a lack of certainty on the variables that drive the park cooling effects and on the multiple roles of trees and greenspaces. Spatial contrasts in nocturnal temperatures between green infrastructure and nearby built-up areas depend on park perimeter and area (Gao et al. 2022; Cai et al. 2023), on proportion of grass and trees, on tree size (Zhu et al. 2021), on vegetation types and arrangements (street trees vs parks), on density of vegetation (Holmer et al 2013), on park



topography (Barradas 1991; Chang et al. 2007), and on local climates (Ibsen et al. 2021). Other
authors investigated the spatial extent of cooling by urban parks, i.e. the *cooling effect*
*distance*, showing that it depends on both park size and park greenness (e.g Zhu et al 2021).
From a recent review of park cooling effect studies conducted by Aram et al. (2019), we
conclude that most studies focus on the impact of park characteristics and investigations on
the impact of meteorological conditions on park cooling effects are rare.

The impact of meteorological conditions, such as cloudiness, wind and turbulence on
differential cooling is studied mostly at regional scale in terms of their impact on the urban
heat island (UHI) intensity (Oke 2017). While the influence of cloud cover and wind is rather
established (e.g. Morris et al. 2001, Lin et al. 2022), also the occurrence and characteristics of
night-time low-level jets are found to influence UHI intensity (Lemonsu et al. 2009; Cespedes
et al. 2024). However, the impact of local- to meso-scale meteorological phenomena on
cooling effects of urban green infrastructure is not well quantified.

The combined effects of green infrastructure characteristics and meteorological regimes on
nocturnal cooling must hence be better understood so that the cooling effect of urban
renaturation projects can be quantified more precisely. Which conditions affect the park
cooling effect intensity? What is the relative impact of park characteristics and meteorological
processes in the urban boundary layer on the cooling intensity ?

The overall objective of this study is to quantify in detail the nocturnal cooling effects of urban
parks during warm summertime conditions, taking into account potential cooling effects from
the rural surroundings. We carried out this study in the framework of the Heat and Health in
Cities project (H2C, Lemonsu et al. 2024) that focuses on the effects of excessive summertime
heat and air pollution on human vulnerability (Forceville et al. 2024) with the Paris region
(France) as a study area. A dedicated field campaign was designed and carried out in the city
of Paris and the surrounding region to monitor spatial and temporal variations in key
atmospheric thermodynamic variables in the urban canopy layer and urban boundary layer
during summer 2022. The measurements performed, including near-surface and vertical
profiles of temperature, humidity, wind and turbulence, and data analysis methodology are
presented in Section 2. Section 3 presents the analysis of urban park cooling effects in relation



to regional UHI and their variability during summer 2022, with a focus on a one-week heat
wave event. Next (Section 4), we investigate the characteristics of the urban boundary layer
structure under three distinct atmospheric turbulence regimes and their influence on park
cooling effects. Finally, we quantify the role of atmospheric stability and vertical turbulent
mixing on differential evening cooling between built-up locations, urban parks and rural
settings (Section 5).

# 116    2 - Data and methodologies

The present study is based upon data collected in the Paris region during the first Special
Observation Period of the Heat and Health in Cities project  (SOP 2022, Figure 1), which was
conducted during summer 2022 (Lemonsu et al., 2024ab). This campaign also benefited from
measurements carried out in the context of other research initiatives such as the Paris 2024
Olympics WMO Research and Development Project (RDP-2024) and the ACTRIS research
infrastructure (Laj et al. 2024). This multi-project context motivated the pooling of resources,
a coordinated strategy for the organisation of the summer-2022 experimental campaigns, and
the development of a joint data repository under the name PANAME (PAris region urbaN
Atmospheric    observations    and    models    for    Multidisciplinary    rEsearch    -    see
https://paname.aeris-data.fr/).

## 128    2.1 Datasets used in the study

This study combines continuous measurements collected from June to August 2022 and 14
one-day intensive observation periods (IOPs), with data collected from mid-June to the end
of July 2022. Measurement locations are shown in Figure 1.

**i)    Surface meteorological stations**

Météo France's operational network consists of some fifty ground-based weather stations in
the Paris region measuring at least air temperature at 2 m AGL with a 6-minute acquisition
time step. A few stations provide additional meteorological parameters such as wind speed
and direction at 10 m AGL, global incoming radiation, precipitation, and cloud cover. The



stations are spread across the region in different areas, but are always installed on the ground
on an open lawn (according to WMO recommendations).

We selected six stations to represent rural settings (Local Climate Zone, Stewart and Oke
2012) of the Paris region (Figure 1), located in Changis, Courdimanche, Fresnoy-La-Riviere,
Maule, Melun, and Pontoise, which is similar to the stations selected by Lemonsu et al. (2015).
The stations are geographically distributed in all directions relative to the city centre of Paris
and located at altitudes ranging 50-90 m above sea level (ASL). In our study, the reference
rural setting conditions of temperature, wind speed and direction are computed as the
average of the variables measured at those six stations (Changis, Courdimanche, Melun, and
Pontoise stations).

Near-surface urban park weather conditions are documented by a Météo-France weather
station located in the Montsouris Park, a 15-ha park located in the 14th district, south of the
Paris city centre. The station, located at an elevation of 75 m ASL, provides 2-m air
temperature and humidity measurements. Wind speed and direction are measured at 25 m
above ground level (AGL). A detailed description of temperature measurements in the
Montsouris Park is provided by Dahech et al. (2020).

The Paris built-up setting conditions are sampled using Internet of Things (IoT) temperature
and humidity measurements. This compact technology opens up new perspectives in
meteorological measurements, particularly in urban environments where measurement and
installation conditions are sometimes complicated. More than twenty IoT stations (DecentLab
DL-SHT35-001 - air temperature and humidity sensor with radiation shield for LoRaWAN) have
been installed in central Paris starting in July 2022. These are compact and lightweight
stations installed on lampposts at a height of approximately 5 m AGL, following the
recommendations made by Oke (2006). The stations have been installed on the north side of
the lampposts to limit sensor warming through solar irradiance. The reference built-up setting
temperature is computed as the average temperature  recorded by four IoT stations located
within 500 m of each other, in the highly urbanised neighbourhood of the Paris Opera House
(hereafter referred to as Opera). Note that these stations were operational only from July 8,
2022. For the period prior to this date (1 June to 7 July), the built-up setting temperature is



derived from the Météo France weather station Lariboisière Hospital (10th district of Paris)
which is located 2 km northeast of the Opera neighbourhood in an equally dense built-up
setting. Comparisons of temperatures measured at Lariboisière and Opera during July and
August 2022 do not reveal any significant differences (not shown). The built-up setting
temperature (at Lariboisière and Opera) is considered not influenced by green space cooling,
as the closest urban park is about 1 km away and cooling effect distances of parks reported
in the literature are far less than 1 km (Aram et al. 2019).

Finally, we used temperature and wind speed and direction measured at the top of the Eiffel
Tower (287 m AGL) to monitor conditions at a height generally located in the nocturnal urban
boundary layer.

**i)    Doppler Wind Lidar**

A Doppler Wind Lidar (DWL) is used in this study to deduce the intensity of vertical turbulent
mixing. The Vaisala DWL WindCube Scan 400 was installed at 90 m above ground level (AGL)
at the top of the Zamansky Tower located on the campus of Sorbonne University in the 5th
district of Paris (QUALAIR atmospheric station location shown on Fig. 1; https://qualair.fr/) to
measure horizontal wind and vertical velocity. In this study, we use  vertical-stare mode of
the DWL to derive vertical velocity variance ($\sigma_{w'}$) profiles. Each variance profile is calculated
from 300 vertical velocity profiles collected during a 5-min period (one profile per second).
Vertical velocity variance profiles are available every 30 minutes. Due to installation setup,
the first gate available for deriving the vertical velocity variance is at 238 m AGL.

**ii)    Windsond**

A Windsond is a lightweight sonde (12 grams) manufactured by Sparv Embedded,  Sweden
(https://sparvembedded.com/products/windsond). This instrument, packaged in a styrofoam
cup, records pressure, temperature, and relative humidity approximately every second.
Latitude and longitude are determined using an onboard GPS receiver. The S1H3 windsond
model calculates wind speed and direction independently from latitude and longitude,
utilising the GPS signal. Thanks to its lightweight design, the balloon size is somewhat



equivalent to a "party balloon", requiring about 50 L of helium, and making it particularly
suitable for probing the lower parts of the troposphere.
For each IOP, three profiles were produced using windsonds to monitor evening cooling at
16, 20 and 00 UTC. The 16 UTC profile corresponds to conditions of maximum daytime
temperatures. The 20 UTC profile samples conditions about 1 hr after sunset, while the 00
UTC profile is performed in conditions close to the maximum nighttime UHI.
Corrections have been applied to raw data as follows. Before the windsond is released, the
temperature and humidity sensors are not ventilated. Unventilated data (before launch) are
thus carefully compared with the first points of the ventilated profile, and corrected if
necessary. As the temperature and humidity sensors are outside the styrofoam cup, the
windsond is subject to the influence of solar radiation during the day. A daytime overheating
on the order of about +1°C was observed by comparing those profiles with data collected by
Vaisala RS41-SGP radiosondes launched at the same time the URBAN-B location (see
Appendix 1). A correction of -1°C was therefore applied across the entire profile for
windsonde data at 16 UTC. No radiative correction is applied at 20 and 00 UTC.

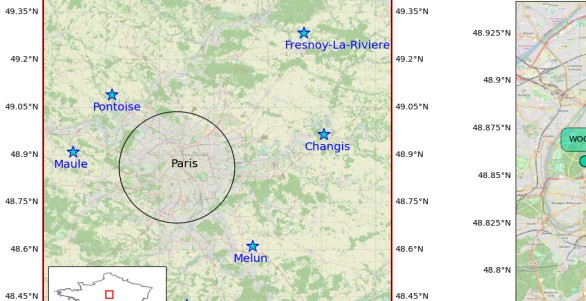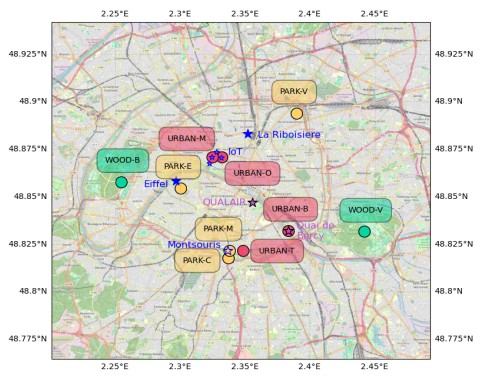

Figure 1: (left) Locations of the six weather stations contributing to the rural setting
reference. (right) Locations of fixed weather stations in Paris city (blue stars) and of
windsond and radiosonde launch sites in urban woods (green dots WOOD), urban
parks (yellow dots PARK) and built-up areas (red dots URBAN). © OpenStreetMap
contributors 2023. Distributed under the Open Data Commons Open Database
License (ODbL) v1.0.





## 2.2 Sampling methodology


Our study focuses on evening temperature evolution at various locations across the Paris
region under predominantly cloud free conditions. The cloud cover fraction is derived on an
hourly basis using a Lufft CHM15k automatic lidar ceilometer located at the SIRTA observatory
(Haeffelin et al. 2005) and a second one located at the QUALAIR atmospheric station. Evening
cloud-free conditions are defined as a cloud fraction less than 20% for each hour between 16
and 00 UTC. In the period June-August 2022, 54 days are classified as "evening cloud-free
conditions".

The 14 intensive observation days were selected to focus predominantly on warm to hot
daytime conditions followed by cloud free nights. Two heat wave events were covered with
intensive observations, the first one on 16-18 June and the second one on 12-19 July.
Windsond launch sites were classified in three types of settings i.e. urban woods, urban parks
and built-up areas. Two urban woods, located East of the city (Bois de Vincennes, 995 ha;
WOOD-V in Fig. 1) and West of the city (Bois de Boulogne, 845 ha; WOOD-B), are mostly
wooded, including open lawns, small lakes, buildings and roads. Three urban parks of
comparable size were selected to sample different neighbourhoods of the city. One is located
south of the city centre (Cité Universitaire about 32 ha with 50% green space and 50%
housings and small roads, located across the street from Montsouris Park; PARK-C), the
second one is West of the city centre (Eiffel Tower park, 24 ha, predominantly trees and open
lawns; PARK-E), and the third one is Northeast of the city centre (La Villette Park, 55 ha
including 30 ha of green space and 25 ha of built-up areas; PARK-V).  Windsonds were also
launched from four built-up areas: one in the 13[th] district close to Montsouris Park (URBAN-T
in Fig 1.), two in the 9[th] district close to the Opera IoT stations (URBAN-M and URBAN-O), and
one in the 12[th] district next to the radiosonde launch site (URBAN-B). For the June 3-day heat
wave, we sampled one park, one wood and one built-up site. For the July heat wave, we were
able to sample the three parks, two woods and two built-up sites. Launch sites are shown in
Fig. 1 and IOP dates and launch locations are shown in Table 1.






**June**

| Mon | Tue | Wed | Thu | Fri | Sat | Sun |
|---|---|---|---|---|---|---|
|  |  | 1 | 2 | 3 | 4 | 5 |
| 6 | 7 | 8 | 9 | 10 | 11 | 12 |
| +RS-B 13 | +RS-B 14 | 15 | +RS-B 16<br>X PARK-C<br>O PARK-C<br>[-] PARK-C | +RS-B 17<br>X WOOD-V<br>O WOOD-V<br>[-] PARK-C | +RS-B 18<br>X URBAN-T<br>O PARK-C<br>[-] URBAN-T | +RS-B 19 |
| 20 | 21 | +RS-B 22 | 23 | 24 | 25 | 26 |
| +RS-B 27 | +RS-B 28<br>X WOOD-V<br>O WOOD-V<br>[-] WOOD-V | +RS-B 29 | 30 |  | X 16:00 UTC<br>O 20:00 UTC<br>[-] 00:00 UTC |  |

**July**

| Mon | Tue | Wed | Thu | Fri | Sat | Sun |
|---|---|---|---|---|---|---|
|  |  |  |  | 1 | +RS-B 2 | +RS-B 3 |
| +RS-B 4<br>X WOOD-B<br>O WOOD-B<br>[-] WOOD-B | +RS-B 5<br>O URBAN-O<br>[-] URBAN-M | +RS-B 6 | 7 | 8 | 9 | 10 |
| 11 | 12<br>X PARK-E<br>O PARK-E<br>[-] PARK-E | 13<br>X WOOD-B<br>O WOOD-B<br>[-] WOOD-B | 14<br>X URBAN-B<br>O URBAN-B | 15<br>X PARK-V<br>O PARK-V<br>[-] PARK-V | 16<br>X PARK-C<br>O PARK-C<br>[-] PARK-C | 17<br>O WOOD-B<br>[-] WOOD-B |
| 18<br>X URBAN-M<br>O URBAN-M<br>[-] URBAN-M | 19<br>X WOOD-V<br>O WOOD-V | 20 | 21 | 22 | 23 | 24 |
| 25 | 26 | 27 | 28 | +RS-B 29 | +RS-B 30 | +RS-B 31 |


Table 1: Dates of the 14 IOP with location and time of launch of windsonds.
Locations are shown in Fig. 1. +RS-B indicates that radiosondes were launched from the
URBAN-B location at the same time as the windsonds. The colour indicates the location type
for each day as Urban Park (Yellow), Urban Wood (Green), or built-up setting (Red).



# 3 - Urban park cooling effect in relation to regional-scale UHI

The cooling effect intensity of an urban park is derived as the temperature difference between
a representative built-up neighbourhood and the green infrastructure where we expect
cooler nocturnal conditions. In our study, the cooling effect intensity of the Montsouris Park
is computed, on an hourly mean basis, as the deficit of temperature measured in the park
relative to the air temperature measured in the built-up setting (at Lariboisière and Opera
sites - see detailed definition of locations in Section 2). As park cooling effect intensity is
reported to be highly variable, we study this variability as a function of the nocturnal UHI in
the Paris region, which represents the regional-scale temperature contrasts between the



same built-up environment and the vegetated rural reference. The study covers summer 2022
focusing on the 54 evening periods with cloud-free conditions (defined in Section 2).

## 3.1 Summertime urban park cooling effect variability

The regional UHI is known to be dependent on both cloud-cover fraction and wind speed.
Here we focus on cloud-free nights, for which the UHI has been found to be proportional to
the inverse of the square-to-third root of the wind speed (e.g. Morris et al. 2001). Cespedes
et al. (2024) has also shown that the strongest UHI intensities are found for very low vertical
velocity variance values, measured above the urban canopy, and that UHI decreases as
vertical velocity variance increases.
Fig. 2 presents median nocturnal cooling intensity of the Montsouris Park (a 15-ha urban park)
against the median nocturnal regional UHI and median vertical velocity variance computed
over the 19-02 UTC time interval for each night. A K-means clustering method based on the
three variables is used to identify different regimes. The figure reveals three different
regimes. In conditions of strongest UHI (6-10°C), we find a group of days where the park
cooling effect intensity ranges 2-5°C. In this regime, the vertical velocity variance is very low
with median nocturnal values ranging from 0.02 to 0.1 $m^2\,s^{-2}$. In these conditions, urban park
cooling intensity relative to the built-up environment shows a strong variability, but is on
average half the regional UHI intensity. In conditions of weak UHI intensity (2-4°C), the park
cooling effect is close to 1°C, while the vertical velocity variances are high (greater than 0.25
$m^2 s^{-2}$). In this regime, intra-urban temperatures are most homogeneous and urban-rural
contrasts are minimal, which is likely due to significant advection. In between, we find a
number of days where the urban park cooling effect remains limited (1-2°C), while the urban-
rural temperature contrasts are significantly stronger (4-8°C), by a factor of about four. In
these conditions, we find that the vertical velocity variances range between 0.1 and 0.2 $m^2\,s^{-}$
$^2$. For those days, the rural environment around the city cools very efficiently, while the urban
setting remains hot with little intra-urban contrasts.
In summary, we can state that:



●  Conditions of strong park cooling intensity combined with strong regional UHI
intensity occur in a regime of low vertical velocity variance. This regime will be referred
to as the stagnant regime in the rest of the paper,

●  Conditions of moderate park cooling intensity combined with strong regional UHI
intensity occur in a regime of moderate vertical velocity variance (referred to as the
intermediary regime).

●  conditions of weak park cooling intensity combined with weak regional UHI intensity
occur in a regime of high vertical velocity variance (referred to as the turbulent
regime).

Based on these findings, several questions arise. What processes drive the evening cooling in
the urban park in these different conditions? What is responsible for the different urban park
cooling effects that we find for low, moderate and high vertical velocity variance?






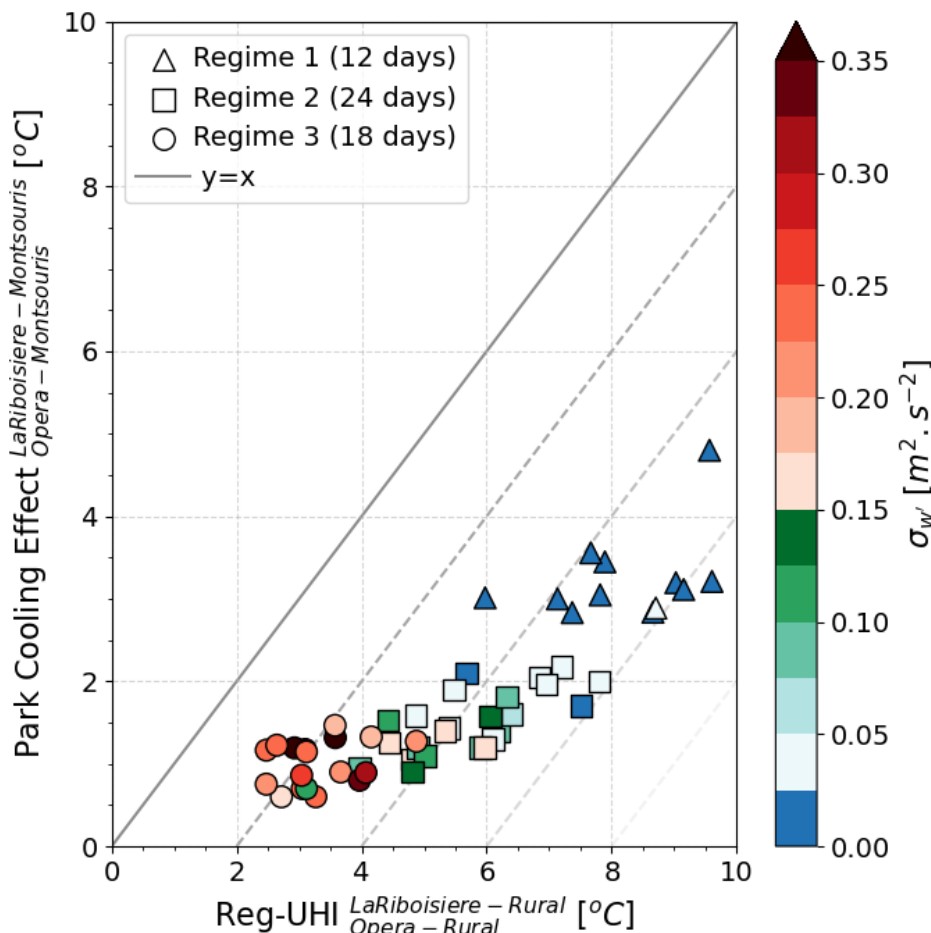


Figure 2. Nocturnal urban park cooling effect intensity against regional-scale UHI intensity and
vertical velocity variance (color scale), derived from 8 hours of measurements (median 19-02
UTC values) for the 54 cloud-free evenings.

## 3.2 Urban park cooling effect variability in a heatwave period


To better understand factors affecting the variability in nocturnal temperature contrasts
between urban parks and the built-up settings, we focus next on an eight-day event (12-19



July 2022) that is characterised by extreme daytime temperature (peak values approaching
40°C on several days) and a set of diverse evening cooling patterns.

This period is characterised by a powerful anticyclonic axis between Morocco, France and the
British Isles, which gradually warmed the air (Fig. 3). A secondary low-pressure system located
between the Azores and Portugal moved towards the Bay of Biscay, strengthening the
advection of particularly hot air from the Iberian Peninsula. This contributed to the
intensification of a heatwave over the European continent, with an extreme peak over the
Paris region on July 19. As it moved north-eastwards over France, this low pressure system
advected cooler oceanic air from the west, causing temperatures to fall and progressing
eastwards with thunderstorm activity. During the heatwave, the 850 hPa temperature
exceeded 20°C, while on standard summer days, it is closer to 10°C.


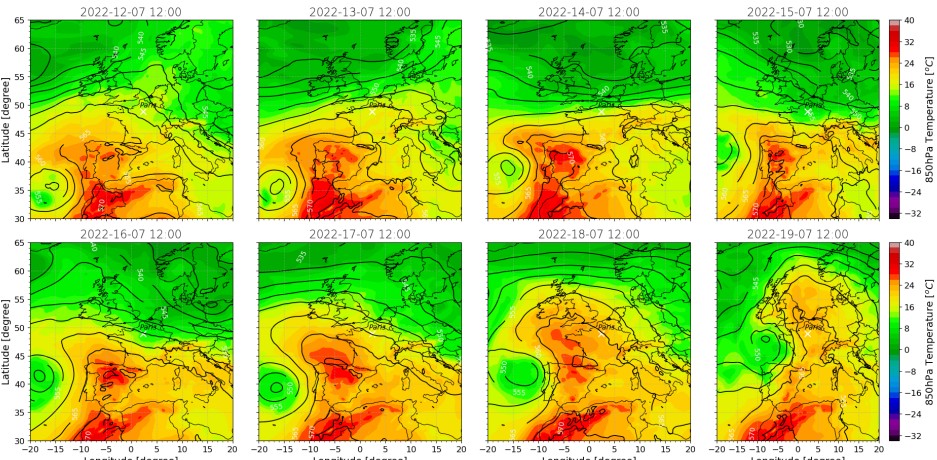


Figure 3: Synoptic overview from 12 to 19 July 2022 based on ERA5 reanalysis. The colour
bar represents temperature at 850 hPa, and the contours represent the geopotential height
difference between 500 hPa and 1000 hPa (dam).

Figure 4 shows the temporal evolution of near-surface atmospheric conditions during the
eight-day period. Figure 4a compares the 2-m air temperature measured in Opera built-up
setting, Montsouris urban park, and the rural reference setting. The regimes identified in
Section 3.1 are also shown for each night. Figure 4b presents the rate of change of



temperature over time at the three locations. Figure 4c shows the temperature differences
between the built-up site and the urban park and the rural setting, respectively. Figure 4d
presents the wind speed and direction measured at the Montsouris urban park 25 m AGL and
Fig. 4e shows the vertical velocity variance measured at 240 m AGL.

The eight-day period is characterised by a first heat wave on July 12 and 13 (stagnant regime),
due to the advection of hot air shown in Fig. 3, with maximum temperature exceeding 35°C,
followed by three days of more moderate heat on July 14, 15, and 16 (intermediary and
turbulent regimes, maximum temperature at or below 30°C and minimal temperatures in the
built-up environment less than 20°C). A second, more intense, advection of hot air occurs the
following three days on July 17, 18, and 19 (stagnant regime) with daytime maximum
temperatures exceeding 35°C. Figure 4a shows that the daytime maximum temperatures
(between 16 and 17 UTC) in the built-up, urban park and rural settings are close, within 1°C
of each other. Conversely, night-time minimal temperatures (between 03 and 04 UTC) differ
by 4-10°C between the built-up and rural settings with significant day to day variations (Figure
4c).

Figure 4b shows positive heating rates from sunrise until about one hour before sunset. Peak
heating rates reach 2-3°C/hr, but are on average near 1°C/hr. One hour before sunset,
temperature changes become negative (cooling). We observe a two-phase cooling consistent
with earlier findings reported in the literature (e.g. Holmer et al. 2013). The first phase lasts
from 16 to 21 UTC. It is characterised by large changes in cooling rate reaching maximum
values near 19-20 UTC and with differences of up to 2°C/hr between built-up, urban park, and
rural cooling rates (on 12/07, 17/07 and 18/07). The second phase starts after 21 UTC and
lasts until sunrise or about 04 UTC. It is characterised by more moderate cooling rates of
typically less than -1°C/hr and by virtually no contrasts between built-up, park and rural
settings.

In the evening, air temperature cooling in the urban canopy is driven by a combination of
processes, including radiative cooling of the surfaces and the air (through radiative flux
divergence), turbulent heat exchange (through sensible and latent heat fluxes), release of
heat from the ground (storage heat flux), vertical mixing of air, and advection (Oke 2017).



These processes are known to depend on the surface types and properties (albedo, emissivity,
heat capacity, soil moisture), the 3-D canopy structure (sky view factor), the city morphology,
anthropogenic heat emissions, the spatial distribution of surface types (urban to rural surface
gradients), and synoptic-scale weather conditions (wind, clouds). According to Steeneveld et
al. (2006), atmospheric static stability and mesoscale dynamics affect the relative contribution
of the radiative and turbulent processes. When the vertical turbulent mixing is low, turbulent
heat fluxes are weak, hence air temperature cooling is dominated by radiative flux divergence,
partially compensated by the storage heat flux.

This is consistent with cooling rates shown in Fig. 4b. In the rural setting and in the urban park,
where the storage heat flux is low, the largest cooling rates (peaking at -3°C/hr and -2°C/hr
respectively) are observed in conditions of low vertical velocity variance (Fig. 4e), on the
evenings of 12/07, 17/07 and 18/07 (stagnant regime). In the built-up area, the radiative
cooling is partially compensated by a stronger ground heat flux. On nights with moderate to
high vertical velocity variance, radiative flux divergence is reduced and also compensated by
sensible and latent heat flux releases, which leads to lower cooling rates in both urban park
and rural settings. The excess of urban-park cooling compared to the built-up environment
lasts four to six hours (from 18 to 00 UTC) as is the case for the rural surface.

The contrasts in cooling rates between the built-up environment, the urban park and the rural
settings can explain the large variability in nocturnal park cooling effect and regional-scale
UHI intensities shown in Fig. 4c. On the three nights with lowest wind speed (<2 m s$^{-1}$, Fig. 4d)
and lowest vertical velocity variance (<0.05 m$^2$ s$^{-2}$), that is on 12-13/07, 17-18/07 and 18-
19/07 (stagnant regime), the maximum regional UHI intensity exceeds 8°C, while the
maximum park cooling effect reaches nearly 4°C. On those nights, in the built-up
environment, the air temperature cools by 7-9°C from sunset to sunrise, while the urban park
cools an extra 3-4°C, and the rural setting an additional 3-4°C. On the night with moderate
wind speed (3-4 m s$^{-1}$) and moderate vertical velocity variance, 15-16/07(intermediary
regime), the regional UHI peaks near 6°C, while the park cooling effect reaches about 2°C. On
this night, the air temperature cools by about 10°C from sunset to sunrise in the built-up
environment, while the urban green infrastructure cools an extra 2°C, and the rural setting an
additional 3-4°C. On the nights of 14-15/07 and 16-17/07 (turbulent regime), the wind speed





exceeds 4 m s$^{-1}$ and the park cooling effect reaches just 1°C, while the maximum regional UHI
intensity is about 4°C.

The analysis of the 12-19 July period confirms the results shown in Fig. 2. Different regimes
exist that influence park cooling effect and regional UHI intensities. In particular, during nights
with very low wind speeds, the air above the urban park cools significantly more (up to 4°C)
than in our reference built-up environment. To better understand the processes and
conditions that affect these nocturnal intra-urban cooling contrasts we will investigate the
dynamics and thermodynamics of the urban boundary layer over green infrastructures of
different sizes in the following section.




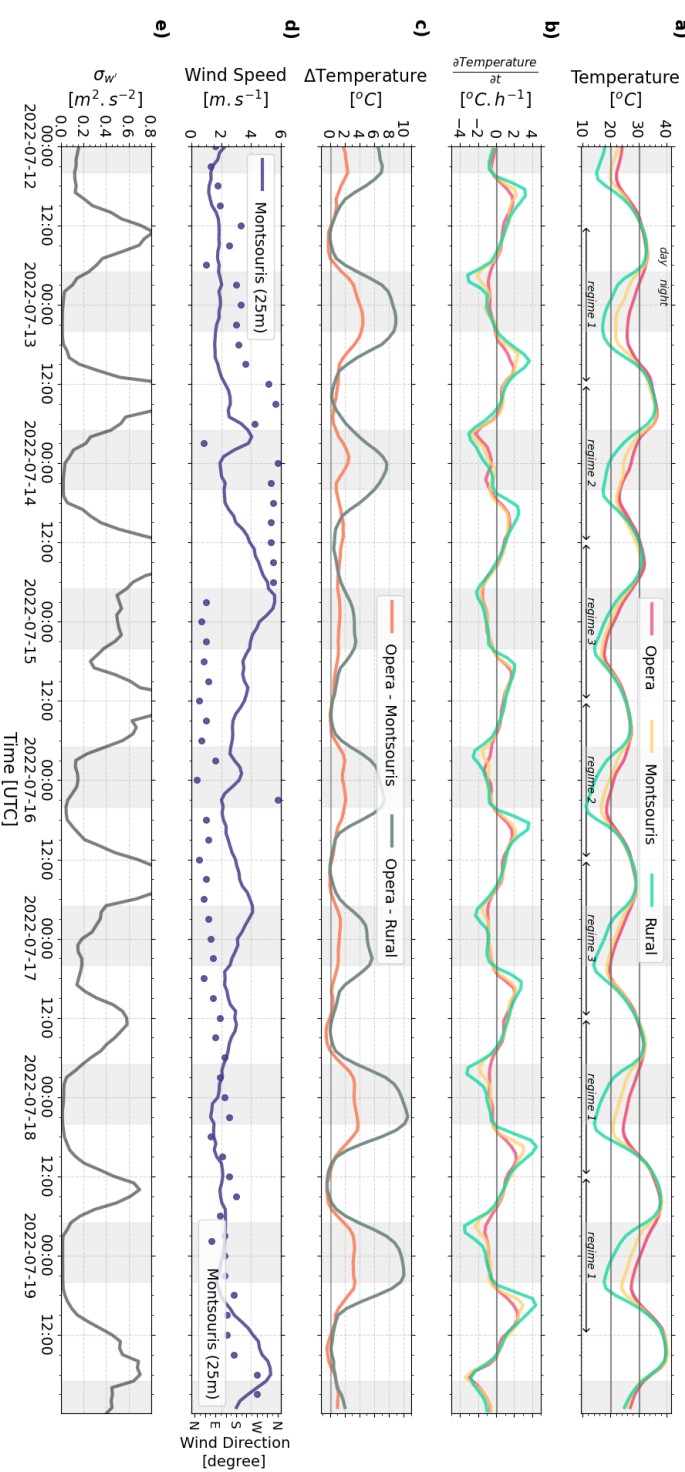



Figure 4: near-surface temperature and nighttime turbulence regimes (a) and cooling rate (b)
measured in a built-up environment (Paris Opera district), an urban park (Montsouris park,
15-ha), and the average of 6 rural locations around Paris; Urban park cooling effect (Opera-
Montsouris temperature difference) and regional-scale UHI (Opera-Rural temperature
difference) (c); wind speed and direction measured at Montsouris Park 25 m AGL (d); and
vertical velocity variance measured in Paris city centre 238 m AGL. 12-19 July 2022; during
that week, the sun sets at about 19 UTC and rises at about 04 UTC.

## 434 4) Evening cooling in and above urban parks and
## 435 urban woods


In this section, we focus on four different nights to study the characteristics of evening cooling
mechanisms above urban green spaces considering dynamics of the urban boundary layer for
the three turbulence regimes. For each evening period (16-00 UTC), we analyse time series of
near-surface temperature, humidity, and wind measured in the built-up environment, urban
green infrastructures, and rural settings. To investigate the relative role of relevant cooling
mechanisms, i.e. radiative cooling of the surfaces, radiative cooling of the air through
radiative flux divergence, turbulent heat exchange, vertical mixing, and advection, it is helpful
to quantify conditions in the urban boundary layer. Therefore, in order to assess the relative
roles of surface-driven and atmospheric-driven processes, the conditions measured at the
surface are complemented by the analysis of the observations at the top of the Eiffel Tower
(287 m AGL), as well as vertical profiles of meteorological variables obtained from windsond
profile measurements.

## 450 4.1 Stagnant regime: strong park cooling effect combined with
## 451 strong UHI intensity






Here we focus on two nights that show the strongest park cooling effect intensity and most
significant UHI intensity, classified as stagnant regime, i.e. 12-13/07 and 17-18/07. Both
selected nights occur in high-pressure synoptic conditions with meso-scale subsidence over
the region. Hot air advection driven by a secondary pressure low located west of the Iberian
Peninsula led to 850 hPa temperatures near 20°C. Both nights are characterised by very warm
conditions over the preceding daytime period with daily maximum air temperatures
exceeding 32°C (see Fig.s 5a, 6a). Strong regional-scale UHI and park cooling intensities are
due to sharp contrasts in peak cooling rates (Fig.s 5b and 6b) between built-up, park and rural
settings that last for 4-6 hours. On both 12/07 and 17/07, an evening cooling (16-00 UTC) of -
5°C, -9°C and -14°C is documented in the built-up, urban park and rural settings, respectively,
as shown in Table 2.

| | 16-00 UTC cumulative temperature change [°C] (average cooling rate [°C/h]) | | |
|---|---|---|---|
| **Regimes** | **Opera** | **Park** | **Rural** |
| **Stagnant Regime : Strong park cooling effect and strong UHI intensities** | -5.1 (-0.6) | -9.1 (-1.1) | -14.0 (-1.8) |
| **Intermediary Regime: Moderate park cooling effect and strong UHI intensities** | -5.9 (-0.7) | -7.6 (-0.9) | -12.6 (-1.6) |
| **Turbulent Regime: Weak park cooling effect and low UHI intensities** | -9.6 (-1.2) | -9.4 (-1.2) | -13.1 (-1.6) |

Table 2: 16-00 UTC cumulative evening temperature change and average cooling rate for the
three turbulence regimes.




The relatively strong cooling rate in the urban park compared to the built-up settings suggests
that the surface-driven processes (i.e. radiative cooling and/or turbulent latent heat fluxes)
are rather efficient on those nights. In comparison, the air temperature at the top of the Eiffel
Tower peaks generally around 18 UTC, i.e. about 2 hr later than near the surface at values 2-
3°C colder than the near-surface air temperature (Fig.s 5a and 6a). After 18 UTC, the air starts
to cool with a rate of around -0.35°C/hr, which is nearly half the value of the near-surface
cooling rate measured in the built-up environment (Fig.s 5b and 6b). Hence, the air at 287 m
AGL is only moderately affected by the processes that cool the air close to the surface. This is
the first evidence of decoupling between the urban canopy layer (UCL) and the air above, and
the decrease in static instability in the urban boundary layer (UBL).

Further evidence of this decoupling due to static stability in the UBL can be found in the wind
speed measurements. Figures 5c and 6c show the time series of wind speed at 10 m AGL at
the Melun rural site, at 25 m AGL in the Montsouris urban park and at 287 m AGL at the Eiffel
Tower, for 12/07 and 17/07, respectively. A comparable temporal evolution of wind speed
can be observed in the evening hours on both days. During the afternoon, the wind speed at
both the urban park and the rural site are consistent (about 2-4 m s$^{-1}$ and within 1-2 m s$^{-1}$ of
each other). After about 18 UTC, the wind speed at 287 m AGL increases rapidly to reach 8-
10 m s$^{-1}$ before 00 UTC, while the rural and urban park wind speed remains low at or below 2
m s$^{-1}$, i.e. often lower than during daytime. This is a second evidence that after sunset,
decoupling conditions occur between the surface layer and the air above.

Figures 5 and 6 g and h show vertical profiles of wind speed and direction derived from
windsond profiles launched at 16, 20 and 00 UTC over an urban park (PARK-E; Fig. 1) on 12/07
and a large urban wood (WOOD-B; Fig. 1) on 17/07, respectively. Both IOPs are characterised
by easterly winds with relatively little wind direction evolution in the evening. During daytime
(16 UTC), the wind speed is moderate (2-4 m s$^{-1}$) in the first 700 m of the atmospheric
boundary layer. The windsonds launched after sunset (near 20 UTC) reveal in both cases low
near-surface wind speed (1.5-2.0 m s$^{-1}$) that gradually increases with height (consistent with
results described in the previous paragraph). A 3 m s$^{-1}$ wind shear can be observed on 17/07
between the surface and 200 m AGL. The wind shear is not as strong on 12/07, possibly



because the profile was measured 45 min earlier than on the other day. This wind shear is a
signature of the stabilisation of the atmosphere that inhibits the vertical transfer of
momentum and hence decouples the air aloft from surface drag effects, allowing the wind
speed to increase aloft (e.g. Barthelemie et al. 1996).

The windsonds launched at 00 UTC reveal even stronger windshear between surface and 200
m AGL, with maximum wind speed of around 6.5 m s$^{-1}$ on both nights near 300 m AGL and a
decreasing wind speed above. This vertical structure is known as a low-level jet (LLJ), a
condition that occurs frequently on summer nights above Paris according to Céspedes et al.
(2024). Their work has shown that very low altitude LLJs are associated with low levels of
turbulence, due to the fact that they form in a statically stable atmosphere that inhibits
mechanically induced turbulence.

To characterise the importance of vertical mixing as a potential means for heat transfer
between the UCL and the nocturnal urban boundary layer, we use Doppler wind lidar
measurements to derive time series of vertical velocity variance (Figs 5d and 6d). During the
convective period of the two IOPs, the vertical velocity variance typically exceeds 0.5 m$^2$ s$^{-2}$. It
then decreases rapidly around sunset. At 20 UTC, the values have dropped to less than 0.05
m$^2$ s$^{-2}$ on both 12/07 and 17/07, and remain very low all night. This confirms the very low
vertical turbulent mixing in the UBL on both nights.

To characterise the role of vertical radiative flux divergence in the atmospheric boundary
layer, and to better understand the relative importance of surface-driven vs atmospheric-
driven processes, we analyse the vertical structure of temperature and its temporal evolution.
In the Eiffel Tower urban park (PARK-E), we find that near-surface temperatures measured by
the windsond on 12/07 are consistent with temperatures recorded by the Montsouris urban
park surface station (yellow circles in Fig. 5a). At 20 UTC, we observe a 1°C temperature
inversion between the surface and 50 m AGL (Fig. 5f). Above the inversion, the temperature
decreases adiabatically by about -1°C/100 m so that the potential temperature is nearly
constant in a statically neutral layer between 50 and 700 m (Fig. 5f). At 00 UTC, the surface-
based inversion has become stronger ($\Delta T_{air}$ = 2.5°C and $\Delta\theta_{air}$ = 3.0°C between the surface and



50 m AGL), and two elevated inversions have formed near 100 and 200 m AGL (Fig. 5f, g), with
$\Delta\theta_{air}$ = 0.5°C followed by a statically stable layer with a +0.2°C/100 m lapse rate (Fig. 5g).

In the urban wood (WOOD-B), near-surface temperatures measured by the windsond on
17/07 are close to temperatures measured in the rural settings (green circles in Fig. 6a). With
3.5°C decrease over 50 m, the surface-based temperature inversion at 20 UTC (Fig. 6e) is
already stronger than the inversion observed at 00 UTC over PARK-E on 12/07. Above the
inversion, the temperature decreases adiabatically (Fig. 6e, f) and the potential temperature
profile confirms that the stable wood UBL is capped by a neutral layer above. At 00 UTC, the
surface-based inversion strengthens and extends aloft ($\Delta T_{air}$ = 5.0°C/100 m ; $\Delta\theta_{air}$ = 6°C/100
m), followed by an elevated inversion near 250 m AGL (Fig. 6f, g). The potential temperature
profile is stable between 100 and 300 m AGL (+1.0°C/100 m) and moderately stable
(+0.2°C/100 m) above (Fig. 6g).

These elevated inversions observed both over the urban park and urban wood could be
formed through localised radiative cooling, subsidence and/or advection of statically stable
rural air that is commonly observed above nocturnal UBL (e.g. Tsiringakis et al. 2022). Elevated
inversions in nocturnal UBLs are simulated and studied extensively in Martilli (2002). The drag
and turbulent kinetic energy production induced by the urban structure increases with
increasing wind speed. Vertical mixing of potential temperature leads to a local minimum of
temperature at the location of maximum turbulence through a negative turbulent heat flux.
According to Martilli (2002), the net result of the vertical turbulent transport is to heat the
layer below the base of the inversion and to cool the inversion layer. Cooling of the inversion
layer (roughly between 200 and 300 m AGL) is clearly seen on the both windsond temperature
profiles measured at 00 UTC.

We can conclude that the conditions of stagnant regime, combining strong park cooling
effects and strong UHI intensities, are associated with a significant surface-based inversion
that leads to the decoupling not only of the rural nocturnal boundary layer from the residual
layer but also between the urban boundary layer and the neutral layer above. The strong
stratification suppresses nearly any turbulent vertical motion so that the UBL height is rather
shallow - even below the top of the Eiffel Tower. As the flow is no longer subject to surface



drag, a regional low-level jet forms that likely advects rural, statically stratified air over the
UBL, which can influence the development of elevated inversions. The strong stratification in
the park internal UBL is the result of cooling dominated by radiative flux divergence due to
low turbulent mixing.


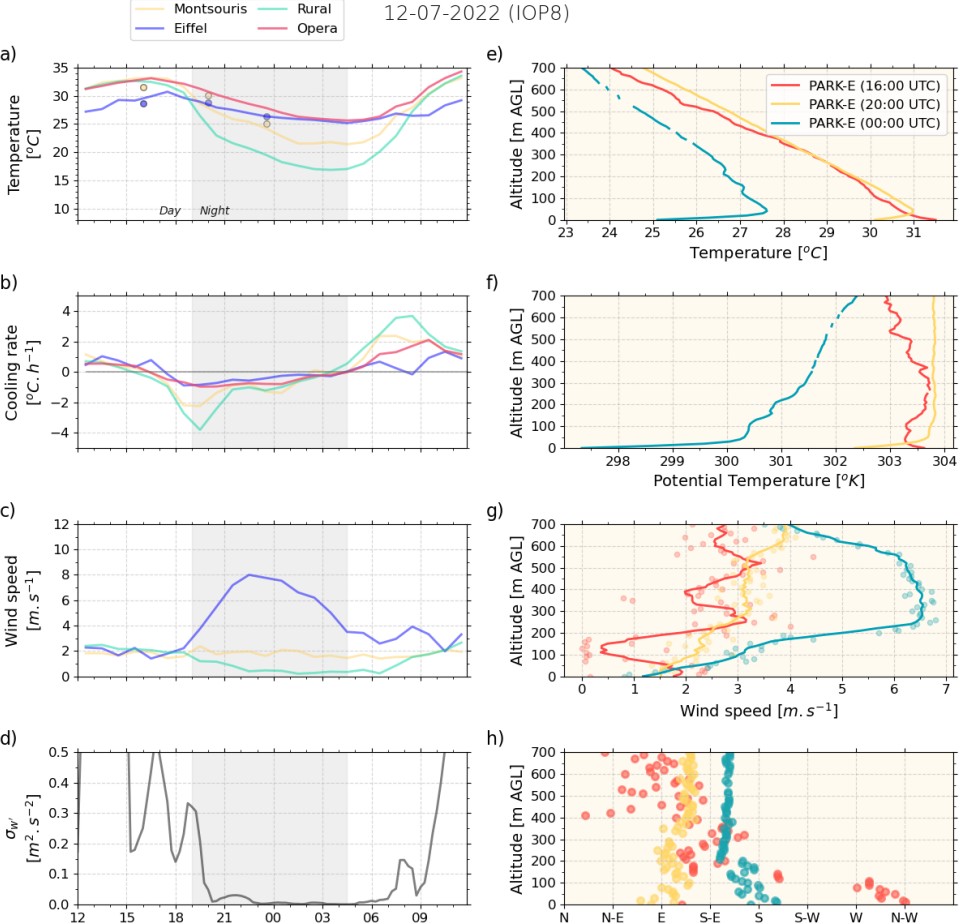


Figure 5: Time series and windsonde profile measurements for July 12, 2022. a-d) Time series
measurements from 12 UTC to 12 UTC (D+1). a) Temperature at Montsouris Park, Rural
settings, Opera (built-up) and top of Eiffel Tower. The coloured dots show the temperature
measured by windsonds at 16, 20, and 00 UTC, respectively at park level and at the height of
the Eiffel Tower (287m AGL). b) Cooling rate at Montsouris, Rural, Opera and Eiffel Tower. c)
Wind speed at Montsouris, Rural, and Eiffel Tower. d) Vertical velocity variance from DWL at



238 m AGL at QUALAIR-SU site. e-h) Vertical profiles from radiosonde measurements released in PARK-E at 16, 20, and 00 UTC, respectively. e) Temperature profile. f) Potential temperature profiles. g) Wind speed profiles. h) Wind direction profiles.

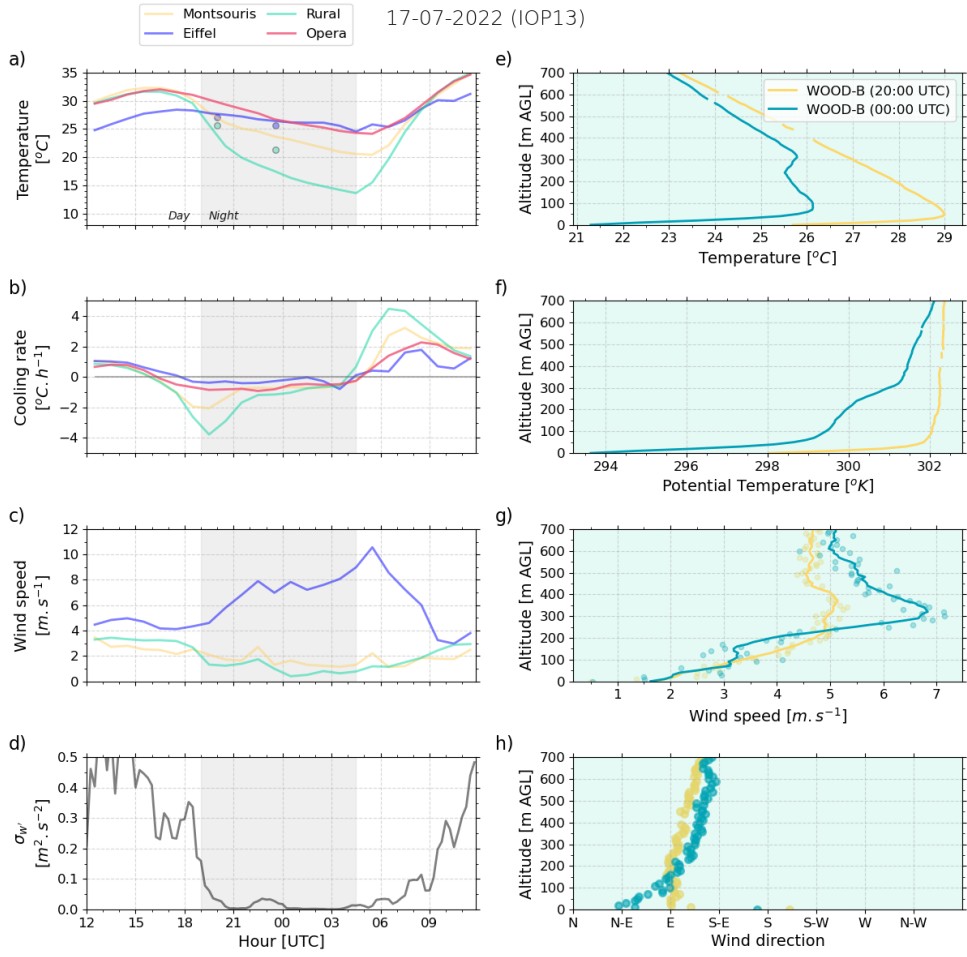

Figure 6: same as Fig. 5 for July 17, 2022.






## 4.2 Intermediary regime: moderate park cooling effect combined with strong UHI intensity


The evening of 15-16/07, compared to those discussed in Section 4.1, is characterised by weaker cooling between 16 and 00 UTC in the rural setting and urban park, and stronger cooling in the built-up environment, as shown in Table 2. It is classified in the intermediary regime. Cooling peaks near -3°C/hr in the rural setting and -1.5°C/hr in urban park, which is slightly less than for the cases of Section 4.1 (Fig. 7b). For this regime, the nocturnal near-surface wind only decreases in the rural setting while it increases in the urban park after 21 UTC as the wind aloft picks up (Fig. 7c) which indicates that vertical momentum transfer is less inhibited above the urban surface. Figure 7d shows that the vertical turbulent mixing remains above 0.1 $m^2 s^{-2}$ after sunset and increases to 0.2 $m^2 s^{-2}$ during the evening which confirms that the UBL remains turbulent during the night.

The windsond profiles carried out in the La Villette urban park (PARK-V on Fig. 1), for which the vegetated area is comparable to that of the Montsouris urban park, reveal at 20 UTC a slight surface-based inversion with a neutral layer above, while at 00 UTC under brisker turbulent mixing the UBL remains near-neutral from the ground up to a temperature inversion near 300 m AGL. It is then likely that the UBL remains neutral due to sensible heat fluxes originating from the hot surface combined with turbulent mixing and from the temperature inversion above. Again, a clear low-level jet with peak horizontal velocity > 9 m $s^{-1}$ near the height of the temperature inversion suggests that stably stratified air from rural surroundings is advected over the city.


The intermediary regime highlights that while the rural nocturnal layer becomes statically stable during the evening, as evidenced by the very low near-surface wind speed at the rural



site, the UBL remains statically neutral. Vertical turbulent mixing in the UBL prevents a
temperature inversion to form in the UCL, even above the urban green space.

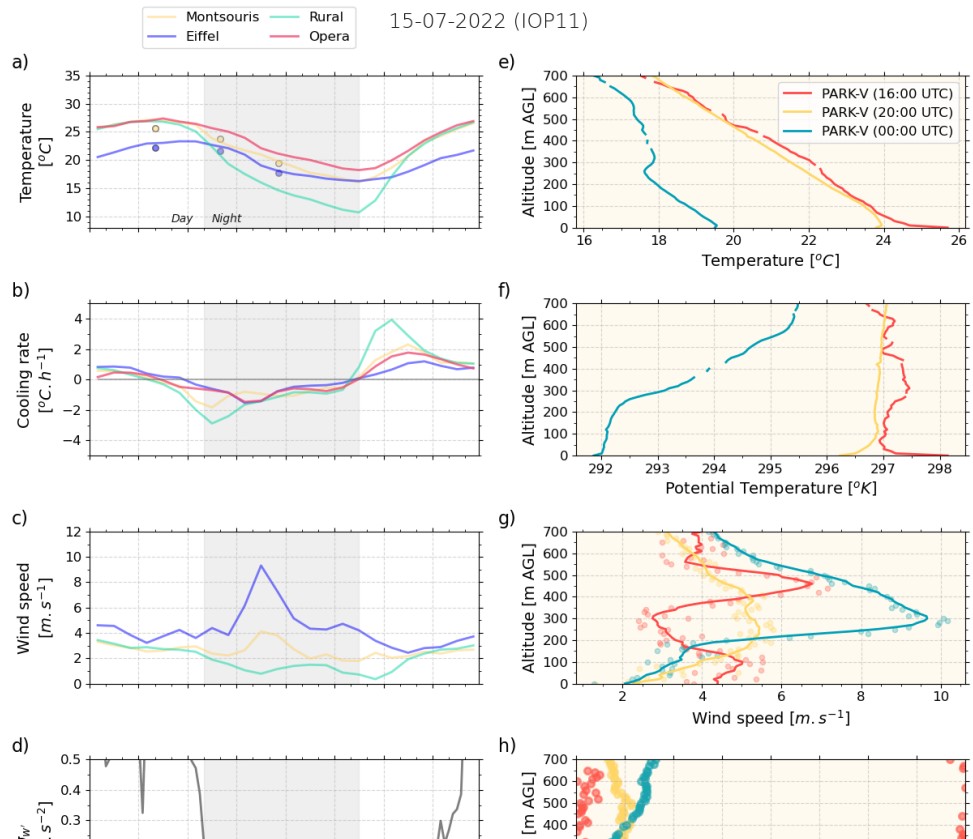


Figure 7: same as Fig. 5 for July 15, 2022 (IOP11)











## 4.3 Turbulent regime: weak park cooling effect combined with weak regional UHI intensity


The evening of 04/07, classified in the turbulent regime, is characterised by nearly identical
cooling rates in built-up settings, urban green spaces, as well as aloft at the top of the Eiffel
Tower. Cooling peaks near -2 to -2.5°C/hr at all locations (Fig. 8b). Wind speed at both the
rural settings and the urban park does not decrease after sunset, but rather increases after
18 UTC as the wind aloft picks up (Fig. 8c). In addition to the strong advection effects, the UBL
remains turbulent during the night as turbulent vertical mixing remains above 0.2 $m^2s^{-2}$ after
sunset (Figure 8d), both indicating that vertical momentum transfer is not inhibited across the
region.

The windsond profiles carried out at the Bois de Boulogne large urban wood (WOOD-B in Fig.
1), detected a neutral UBL from 0 to 700 m AGL at 20 UTC. At 00 UTC, under continued brisk
turbulent mixing, a weak 1°C temperature inversion forms over the large green space while
the neutral UBL extends from 100m to 600 m AGL and is capped by a 5°C temperature
inversion.



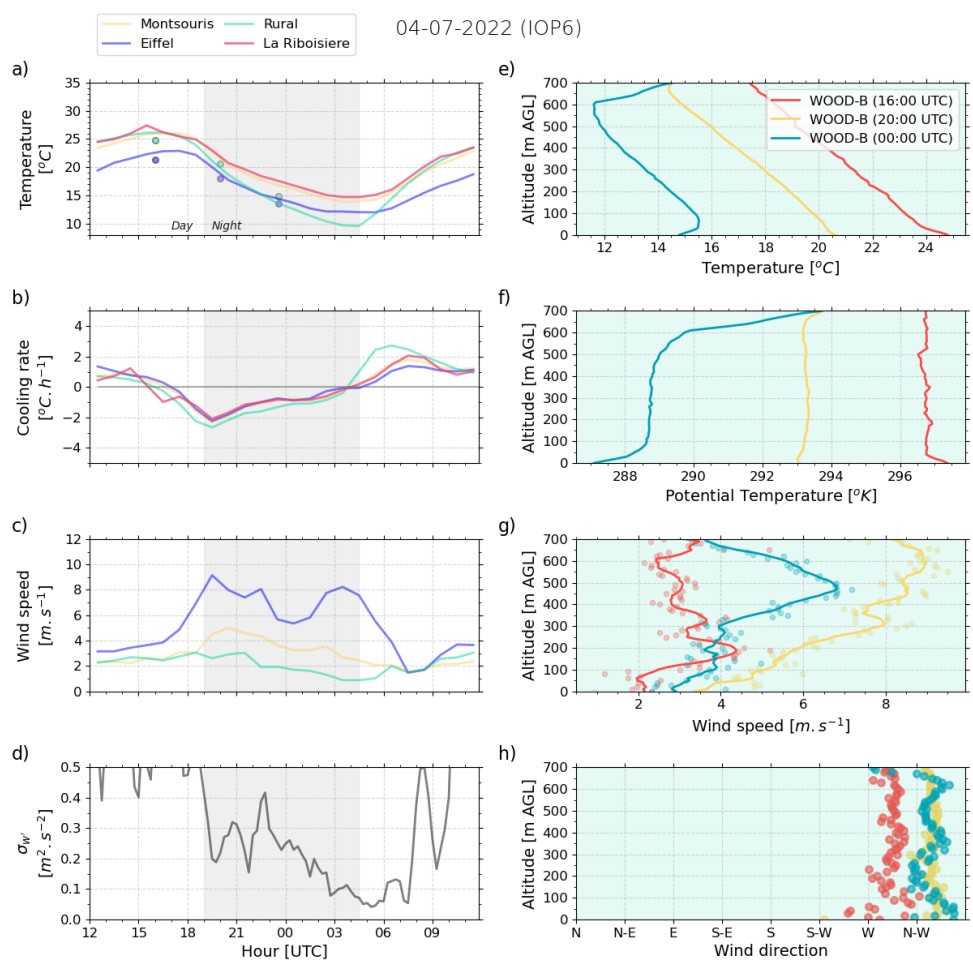


Figure 8: same as Fig. 5 for July 04, 2022













# 5 - Characteristics and impacts of turbulence regimes

To better understand the impact of wind, turbulence and static stability on differential cooling between built-up areas, urban parks and rural settings, we analyse the characteristics of the three turbulence regimes encountered during summer 2022. First, we study the diurnal evolution of wind and turbulence in built-up settings, urban green spaces and rural surroundings (Section 5.1) and then investigate the atmospheric static stability in the built-up surfaces and green infrastructures (Section 5.2) for the three regimes. Finally, we analyse the diurnal cycle of temperature and discuss the nocturnal cooling in built-up environments, green infrastructures and rural settings for the three regimes (Section 5.3).

## 5.1 Wind and turbulent mixing characteristics of turbulence regimes

First, we study how wind speed evolves at diurnal scales over the city (Montsouris urban park), in the rural setting (Melun), and aloft (top of Eiffel Tower) for the turbulence regimes identified in Section 3 (Fig. 9).

In the stagnant regime (highest UHI intensity and lowest vertical velocity variance), we find that at sunset, when vertical mixing drops, the wind speed aloft increases while the near-surface wind speed decreases both over the urban park and in the rural setting (Fig. 9a). Vertical velocity variance reaches values below 0.05 $m^2 s^{-2}$ shortly after sunset. Not only the rural nocturnal boundary layer but also the UBL becomes stratified, thereby inhibiting vertical transfer of momentum. The stable UBL becomes decoupled from the neutral layer above, allowing near-surface wind speeds to decrease, on average below 2 m $s^{-1}$, through surface drag, while wind speed aloft experiences reduced friction and hence increases.

In the intermediate regime (strong ΔUHI and moderate vertical velocity variance), we observe that on average, the vertical velocity variance decreases later than in the stagnant regime and it is 50 % stronger at sunset, reaching 0.15 $m^2 s^{-2}$ on average during the night (Fig. 9b). The



near-surface wind speed in the rural setting decreases at sunset similarly to the stagnant
regime, so we can hypothesise that the atmosphere becomes stable in the rural environment.
In the urban green spaces, the near-surface wind speed remains unchanged after sunset,
which is consistent with a continued vertical transfer of momentum. Still, the stable
stratification over the rural area tends to favour the formation of a low-level jet, a
phenomenon that occurs in Paris in 70% of the nights in summer 2022 (Cespedes et al. 2024),
so that the wind speed above the neutral UBL can double in magnitude between noon and
midnight.

In the turbulent regime (low UHI intensity and high vertical velocity variance) vertical velocity
variance in the UBL is on average above 0.3 $m^2\,s^{-2}$ at sunset (Fig. 9c). Near-surface wind speed
in the rural setting remains above 3 m $s^{-1}$ on average, while central urban wind speeds
increase consistently across the UBL, i.e. both near the surface and at the top of the Eiffel
Tower.



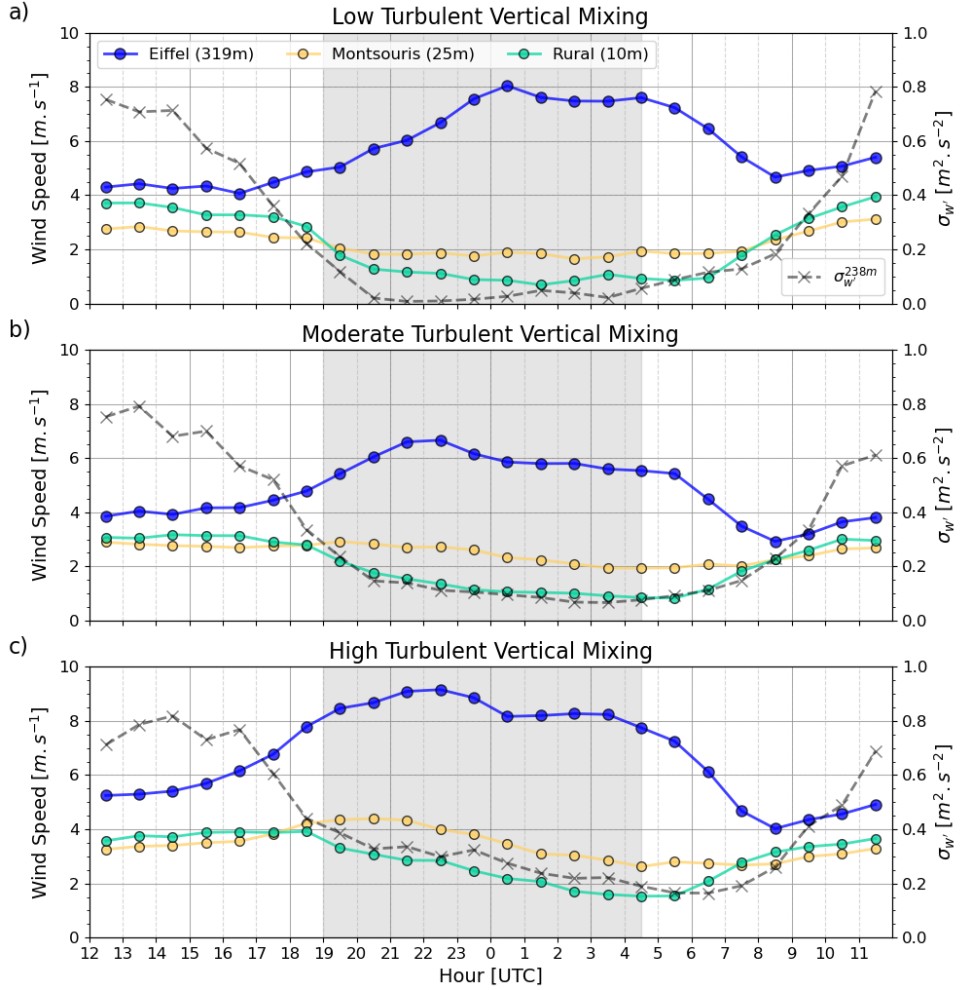

Figure 9 : Average diurnal cycles over summer 2022 for each of the turbulence regimes (stagnant at the top, intermediary in the middle, and turbulent at the bottom): wind speed measured at Melun (rural site); Montsouris park (urban park); top of Eiffel Tower; and vertical velocity variance at 238 m AGL derived from Doppler Lidar measurements.

## 5.2 Atmospheric stability characteristics of turbulence regimes

In Section 4, we found evidence that the static stability above urban parks and urban woods can vary significantly depending on the turbulent vertical mixing in the UBL. To study this variability, we derive the potential temperature lapse rates for each windsond profile carried





out at 20 and 00 UTC above urban woods and parks, as well as radiosonde profiles launched
at the same time from the built-up area of Bercy (URBAN-B location on Fig. 1) along the Seine
river, and plot them against the vertical velocity variance estimated from the DWL
measurements at the same time (Fig. 10). The potential temperature lapse rate is derived for
two vertical intervals, 0-50 m AGL representing the height over which surface-based
inversions are typically observed (also called park/wood internal boundary layer), and 100-
200 m AGL representing the nocturnal UBL. Vertical velocity variances shown in Fig. 10 are
one-hour average values. The turbulence regime derived for each evening (19-02 UTC) is also
shown. Fig. 10b reveals that, when the vertical velocity variance drops below 0.05 $m^2\,s^{-2}$
(corresponding mostly to the stagnant regime) the near-surface potential temperature lapse
rate above urban parks (about 20 ha) ranges 4-6°C/100 m while those above the woods (about
900 ha) can reach 8-14°C/100 m. In the lowest vertical velocity variance conditions (< 0.025
$m^2\,s^{-2}$), near-surface potential temperature lapse rates in built-up areas also become positive
ranging 1-3°C/100 m. This confirms that stable stratification can occur in all settings, but the
strength of the stratification depends on the surface type.
For vertical velocity variances ranging 0.1-0.2 $m^2\,s^{-2}$, near-surface potential temperature lapse
rates above parks and woods range between 0-3°C/100 m, decreasing to near adiabatic
conditions (0°C/100 m) as turbulent mixing increases. In built-up areas, we find that near-
surface potential temperature lapse rates become negative (near -1°C/100 m) as soon as the
vertical velocity variance exceeds 0.05 $m^2\,s^{-2}$, a signature of a typical unstable urban surface
layer.
This analysis provides quantitative evidence that evening and night-time air temperature
conditions in the UCL become spatially heterogeneous when turbulent mixing in the UBL is
very weak. Only then it is possible for a strong temperature inversion to form over the urban
green space through the support of radiative flux divergence. The cool air remains in a local,
internal park/wood thermal boundary layer and does not mix with the relatively warm air in
the surrounding neighbourhoods. The significance and vertical extent of this cool air pool
increases with green space size, and it can be speculated that also green fraction and soil
moisture levels would enhance the effect.

The turbulent mixing in the UBL varies with the static stability of the UBL. As shown in Fig.
10a, when the potential temperature lapse rate at 100-200 m AGL increases to values near



+0.5°C/100m for all settings, including built-up areas, the vertical velocity variance decreases
below 0.05 m$^2$ s$^{-2}$. No clear contrast in stability is found above the different surfaces,
confirming that under the stagnant regime, the nighttime UBL is very shallow.

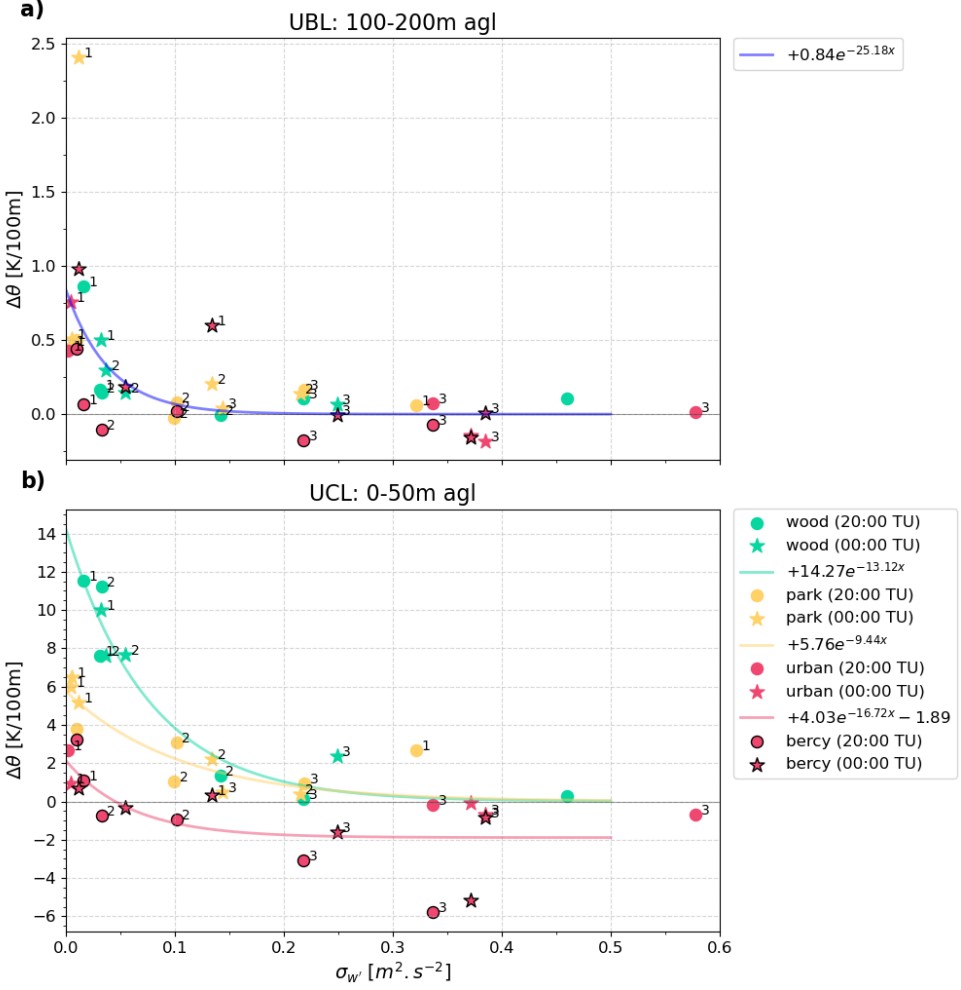




Figure 10: Nighttime (20 and 00 UTC) potential temperature lapse rate above wood (green),
park (orange) and built-up areas (red) as a function of $\sigma_w$ in the UBL (at 240 m AGL) for (a) a
layer between 100-200 m AGL and (b) a layer between 0-50 m AGL. Symbols indicate time
UTC. Urban labels with black borders correspond to data from radiosoundings launched from
the URBAN-B site and the others to data from windsonds (various sites). The number shows
the mean evening (19-02 UTC) turbulence regime for each case.

## 5.3 Impact of turbulence regimes on diurnal temperature
## evolution

Ultimately, we want to determine how the turbulence regimes can impact the nocturnal
cooling provided by urban green infrastructures. Figure 11 shows the mean diurnal cycles of
temperature for stagnant, intermediary and turbulent regimes (a, b, and c, respectively). The
temperature diurnal cycles are normalised by subtracting the temperature measured at 16
UTC (peak daytime temperature). On average, daytime peak temperatures are highest for the
stagnant regime near 31°C, while they peak at about 27°C for the other two regimes. Figure
11 shows that after 16 UTC, the temperature at all sites decreases to reach a minimal value
the next morning at sunrise. In 12 hours, the temperatures drop between 8 and more than
14°C depending on the surface type and the turbulence regime. The stagnant regime reveals
the strongest contrasts between the settings (Fig. 11a). At 00 UTC, five hours after sunset, the
built-up neighbourhood cooled by 5.5°C, while the urban park cooled by 9.0°C and the rural
sites by almost 13.8°C. This confirms earlier findings (Table 2 and Section 5.2) that under low
turbulent vertical mixing, the radiative cooling of the surface in urban park and rural settings
combined with low turbulent vertical mixing provides an efficient cooling of the near-surface
atmosphere. In such conditions, urban parks can provide significantly cooler conditions than
the built-up neighbourhoods nearby.
In the intermediary regime, the evening cooling rate in the built-up environment is slightly
larger than for the stagnant regime (-6.2°C at 00 UTC, Fig. 11b). In the urban park, the
increased UBL turbulent vertical mixing reduces the strength of the near-surface radiative flux
divergence. The evening cooling in the urban park is not as strong (-7.5°C at 00 UTC) as in the



stagnant regime. In the rural setting, the evening cooling is also reduced in the intermediary
regime (-11.7°C at 00 UTC) compared to the stagnant regime, revealing that turbulence is also
likely stronger in the rural nocturnal boundary layer.
In the turbulent regime, with stronger turbulent vertical mixing and higher near-surface wind
speed than in the other regimes, the efficiency of the surface-driven cooling in the rural
setting is even more reduced, which limits the cooling compared to less turbulent conditions
(-10.6°C at 00 UTC, Fig. 11c). In the built-up environment, the air temperature drops by 7.6°C
between 16 and 00 UTC, i.e. 1.5-2°C more than in the other regimes. In this turbulent regime,
the city centre benefits from the cooling of the rural surroundings through advection – the
cooler air is mixed down into the UBL. In the urban park, two competing processes occur. The
radiative flux divergence is reduced by the strong mixing, but this again means cooler air
advected from rural surroundings is efficiently mixed down thereby contributing to a strong
cooling also in the urban park. Hence, we find that the temperature drops by 8.3°C on average
between 16 and 00 UTC, which is in between the stagnant and intermediary regime cooling.






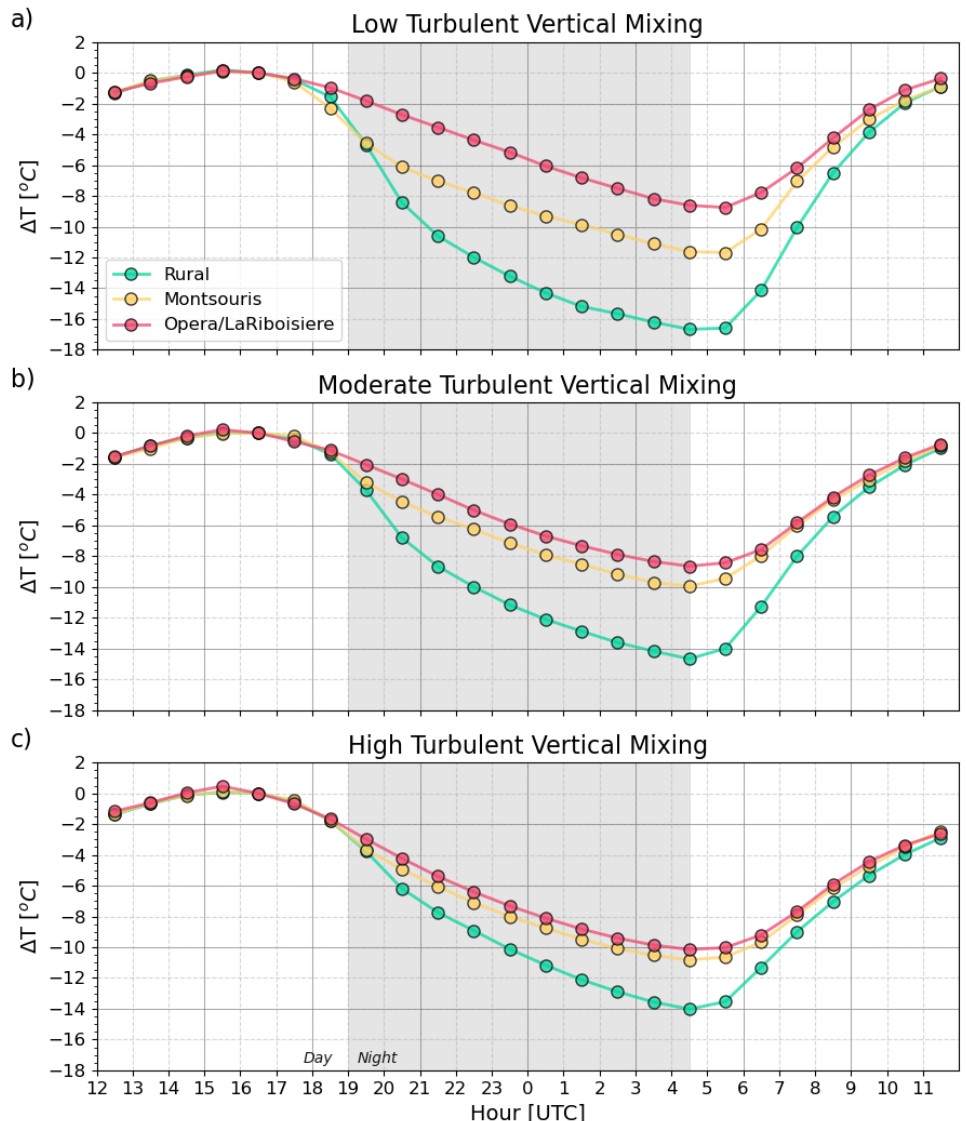

Figure 11: diurnal cycle of temperature difference relative to the temperature at 16 UTC at Melun (rural site), Montsouris park (urban park) and Opera/Lariboisiere (Built-up setting) for stagnant, intermediary and turbulent regimes.



# 6) Conclusions

This study shows that the nocturnal cooling effect of urban parks depends on their characteristics, such as their size, but also on UBL turbulent mixing and static stability regimes that drive the relative importance of radiative and mixing transport cooling processes in the UCL. We find that turbulent vertical mixing conditions measured by a Doppler Lidar at about 240 m AGL in the city centre is a very useful indicator to distinguish different evening cooling regimes in the urban environment.

Highest green space nocturnal cooling intensity occurs under stable stratification in the UBL (statically stable, low turbulent mixing: vertical velocity variance of less than 0.05 $m^2 s^{-2}$) over both rural settings and urban parks. This stagnant regime is associated with large-scale subsidence and large-scale advection of warm air aloft. The potential temperature profiles above the urban parks and woods become statically stable soon after sunset due to radiative cooling of the surface and subsequent cooling of the air by radiative flux divergence, in the absence of a significant turbulent heat flux. A few hours after sunset, the entire UBL becomes on average statically stable (about 200-300 m deep) due to subsidence and advection of the stable rural air above the urban environment. Even if the heat release from the urban surface would in theory lead to an unstable/near neutral urban boundary layer at night, we observe that the strong stabilisation from above limits it strongly in height, or even totally inhibits it. At the top of the UBL, a low-level jet develops over the night with peak wind speed, but mechanical turbulence is inhibited by the static stability of the UBL. The advected rural air mass remains stable above the urban environment because of unusually low vertical mixing conditions. This stagnant regime exhibits the strongest evening cooling in both rural settings and urban parks, and the weakest cooling in the built-up environment, hence strong nocturnal temperature contrasts occur in the city depending on the vegetation fraction. In this regime, the cooling effect of green infrastructure will depend on their size and likely on the vegetation fraction of these areas. In this stagnant regime, we find comparable nocturnal cooling rates (peaking at -2°C/hr around sunset) and static stability in the UCL (lapse rate near 6°C/100m at 00 UTC) above the Montsouris park (15 ha) and the Eiffel tower park (24 ha) that are roughly of the same size.



A second regime is identified, characterised by moderate turbulent vertical mixing in the UBL
(for vertical velocity variance between 0.1 and 0.2 $m^2\,s^{-2}$). Under this intermediary regime, the
potential temperature profiles above the urban park become neutral after sunset. A small
temperature inversion (<0.5°C) can be found in the UCL. A few hours after sunset, the UBL
remains statically neutral up to 200-300 m due to positive turbulent heat flux at the surface
and at the top of the UBL which is characterised by a temperature inversion. Advection of
rural air brings a statically stable layer above the UBL. Under this intermediary regime, the
evening cooling in rural settings is about 2°C less than in the stagnant regime. Two hours after
sunset, the cooling in the urban park is also 2°C less than in the stagnant regime, while the
built-up environment is slightly cooler. There is probably vertical and also horizontal air mixing
(advection or local turbulence), which diminishes the cooling effect of small to medium-sized
parks (15-25ha) by mixing air from surrounding dense neighbourhoods. Hence in the
intermediary regime the intra-urban temperature contrasts between areas with varying
vegetation fractions are significantly reduced.

The third regime identified in this study results in the weakest nocturnal temperature
contrasts. Compared to the stagnant and intermediary regimes, the turbulent regime is
characterised by stronger advection and mesoscale circulation, wind shear and turbulent
vertical mixing. The UBL above the urban park becomes neutral after sunset, with a depth
that is significantly increased (>600 m) compared to the two other regimes. The UBL remains
neutral even several hours after sunset. In this regime, the evening cooling rates are nearly
identical in the built-up environment and in the urban parks. In the turbulent regime, high
turbulence and wind mix the air and homogenise temperatures at a larger scale (district-to-
city scale) than in the intermediary regime (neighbourhood scale), completely encompassing
and erasing the cooling effect of parks.

As statically stable low turbulent mixing conditions occur during the strongest heat waves due
to large-scale subsidence and advection of hot air, it is important to maintain spatially
distributed and accessible vegetated cool island spots in the city so that people can benefit
from cooler outdoor night-time conditions after being exposed to significant daytime heat
stress.



# Data availability

All raw data are available from the AERIS data centre catalogue at https://paname.aeris-data.fr/data-catalogue-2/.

# Author contributions

MH, SK, AL, and VM planned the campaign; MH, SK, JFR, JCD and JC performed the measurements; MH, JFR, SK and JC analysed the data; MH and SK wrote the manuscript draft; JFR produced the figures; AL, VM and TN reviewed and edited the manuscript.

# Competing interests

The authors declare that they have no conflict of interest.

# Acknowledgements

The PANAME experimental program benefited from several supports, including the research project H2C funded by the French national agency for research (ANR) with the reference ANR-20-CE22-0013, the Research Demonstration Project for Paris Olympics 2024 funded by Météo-France and the Weather Meteorological Organization, the Paris Region PhD program 2020, investments from DIM QI2, OBS4CLIM-PIA3, CNRS-INSU and the ACTRIS research infrastructure, and data management (AERIS national data and services center). The authors would like to thank all the volunteers and participants who contributed to the success of the SOP 2022, in particular the teams and many volunteers who carried out the windsonde releases in the parks in the evening and at night and the radiosoundings at Bercy. Thanks are extended to Hugo Ricketts for training the IPSL team to operate windsonds, with the support of the European COST action PROBE. Authors would like to express their thanks to the QUALAIR-SU scientific team who enabled the Doppler Lidar deployment on the site at



Sorbonne Université.

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



**Appendix A:** Windsond temperature profiles
assessment

The evaluation of the Windsond temperature profiles was conducted by comparing them with
the Vaisala RS41 temperature profiles launched at Quai de Bercy (URBAN-B site in Fig. 1)
during the SOP 2022. Data from seven IOPs were used for this evaluation, with profiles
recorded at 16:00, 20:00, and 00:00 UTC, respectively. Von Rohden et al (2022) find a
radiation bias of 0.1°C in Vaisala RS41 temperature data in the troposphere. Our comparisons
reveal an average warm bias of 1.2°C in windsond temperature profiles compared to Vaisala
RS41 values of 16 UTC profiles. No significant bias is found at 20 and 00 UTC.

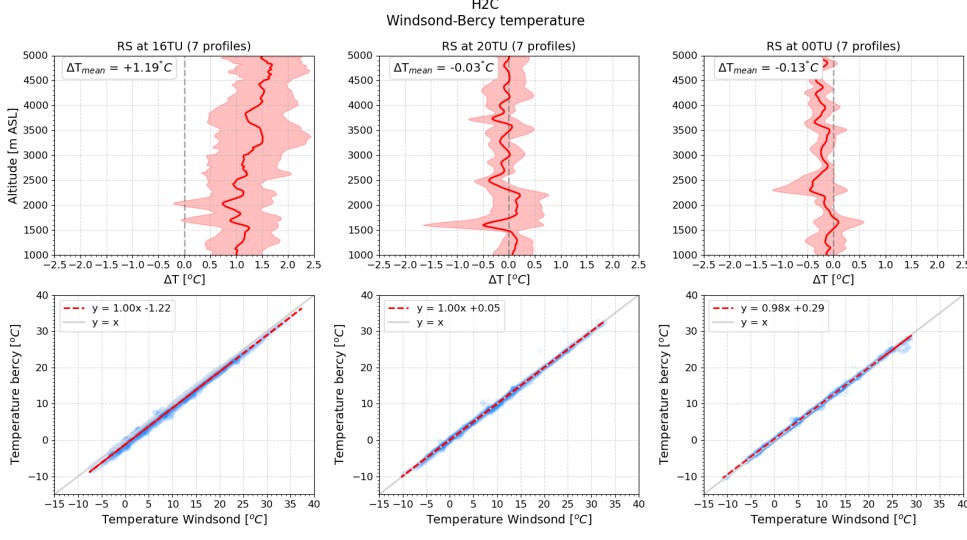


Figure A1: Assessment of windsond temperature profiles. a-c) Average temperature
differences between the Windsond and Vaisala RS-41 temperature profiles from 1000m to
5000m ASL at 16, 20 and 00 UTC respectively. d-f) Point-to-point correlations between
Windsond and Vaisala RS-41 temperatures.

