# Peer review of "Impact of boundary layer stability on urban park"

_EGUsphere, 2024_

## Referee Comment (RC1)

This manuscript used meteorological surface station data and compact radiosondes to examine how the cooling effect of urban parks at night changes with park size and weather conditions in hot summer evenings. The findings are interesting. However, this study should address the following concerns before publication.

1、Major concern: The findings of this study are very interesting and the evidence is very solid. However, it is recommended that the author should create a schematic diagram for each of the three different regimes, placing them in the conclusion section. This will not only help the author to refine the results but also help readers to understand the conclusions of this study more intuitively.

2 、 Suggestion: If possible, in the future, it is recommended that the authors collect observational data from other cities to analyze whether cities with different terrains and climates also reach similar conclusions. This would be a very meaningful study.

3、line 25: "near-by" should be "nearby"

4、line 61: "Trees can provide efficient shading whereby reducing daytime air temperatures" should be "Trees can provide efficient shading, reducing daytime air temperatures"

5、line 65: "The green infrastructures" should be "Green infrastructures"

6、line 79: "e.g" should be "e.g."

7、line 190: "Due to installation setup" should be "Due to the installation setup"

8、line 239: "i.e." should be "i.e.,"

9、line 332-333: "the advection" should be "advection"

10、line 335: "July 19" should be "19 July"

11、line466: "three turbulence regimes" should be "three types of turbulence regimes"

12、line 610-611: "sensible heat fluxes" should be "sensible heat flux"

13、line 768: "the temperature at all sites decreases to reach a minimal value the next morning at sunrise." Should be "the temperature at all sites decreases to a minimal value by the next morning at sunrise."

14、line 811: "is" should be "are"

15、line 848: "Hence in the" should be "Hence, in the"

---

## Author Comment (AC1)

**EGUSPHERE-2024-1777**
**RC1 review**

The authors would like to acknowledge the time and effort spent by the reviewer RC1 on our manuscript. We provide replies to each reviewer comment.

RC1 – General comment:

This manuscript used meteorological surface station data and compact radiosondes to examine how the cooling effect of urban parks at night changes with park size and weather conditions in hot summer evenings. The findings are interesting. However, this study should address the following concerns before publication.

RC1 – Comment 1:

1、 Major concern: The findings of this study are very interesting and the evidence is very solid. However, it is recommended that the author should create a schematic diagram for each of the three different regimes, placing them in the conclusion section. This will not only help the author to refine the results but also help readers to understand the conclusions of this study more intuitively.

Author Reply to comment 1:

We fully agree with the reviewer that a schematic diagram would help to illustrate the main conclusions associated with the three turbulence regimes. We are working now on establishing such a schematic that will represent the nocturnal atmospheric stability in the UBL above built-up environments, urban parks and urban woods using a schematic representation of the potential temperature profile from the ground to the residual layer. The schematic will also show the typical wind profile and the thermodynamic processes that impact the nocturnal evolution of the temperature near the ground and at the top of the UBL. One schematic for each turbulence regime. We will post the proposed schematic as soon as it is ready.

RC1 – Comment 2:

2 、 Suggestion: If possible, in the future, it is recommended that the authors collect observational data from other cities to analyze whether cities with different terrains and climates also reach similar conclusions. This would be a very meaningful study.

Author Reply to comment 2:

Indeed, it will be interesting to study the occurrence of turbulence regimes in other cities with different geographical settings and climate conditions. Our next study is to refine our results with more parks to study the impacts of park dimensions, locations, taking into account when possible soil moisture, vegetation types …

RC1 – Comments 3:

Author Reply to comment 3:

All suggested changes have been implemented in the manuscript.

3、line 25: "near-by" should be "nearby"
4、line 61: "Trees can provide efficient shading whereby reducing daytime air temperatures"
should be "Trees can provide efficient shading, reducing daytime air temperatures"
5、line 65: "The green infrastructures" should be "Green infrastructures"
6、line 79: "e.g" should be "e.g."
7、line 190: "Due to installation setup" should be "Due to the installation setup"
8、line 239: "i.e." should be "i.e.,
"
9、line 332-333: "the advection" should be "advection"
10、line 335: "July 19" should be "19 July"
11、line466:"three turbulence regimes" should be "three types of turbulence regimes"
12、line 610-611: "sensible heat fluxes" should be "sensible heat flux"
13、line 768: "the temperature at all sites decreases to reach a minimal value the next morning
at sunrise." Should be "the temperature at all sites decreases to a minimal value by the next morning at sunrise."
14、line 811: "is" should be "are"
15、line 848: "Hence in the" should be "Hence, in the"

---

## Author Response (AR1)

**EGUSPHERE-2024-1777 – REPLY TO REVIEWER COMMENTS**

**RC1 review**

The authors would like to acknowledge the time and effort spent by the reviewer RC1 on our manuscript. We provide replies to each reviewer comment below. All replies and revised text are shown in blue in this document.

RC1 – General comment:

This manuscript used meteorological surface station data and compact radiosondes to examine how the cooling effect of urban parks at night changes with park size and weather conditions in hot summer evenings. The findings are interesting. However, this study should address the following concerns before publication.

RC1 – Comment 1:

1、 Major concern: The findings of this study are very interesting and the evidence is very solid. However, it is recommended that the author should create a schematic diagram for each of the three different regimes, placing them in the conclusion section. This will not only help the author to refine the results but also help readers to understand the conclusions of this study more intuitively.

Author Reply to comment 1:

We fully agree with the reviewer that a schematic diagram would help to illustrate the main conclusions associated with the three turbulence regimes. We prepared such a schematic that represent the thermodynamic conditions in the nocturnal UBL for the each of the three turbulent mixing regimes. The schematic shows the typical nocturnal wind profile and the typical potential temperature profiles above built-up environments, urban parks and urban woods, highlighting different states of atmospheric stability. The schematic also shows the thermodynamic processes that impact the nocturnal evolution of the temperature near the ground and at the top of the UBL.

This figure is inserted in the manuscript as Fig. 12. It is presented at the beginning of the Conclusions Section with the following sentence:

Line 833: "These findings are summarised on a schematic (Fig. 12) that represents, for each turbulent mixing regime, typical night-time vertical profiles of wind and potential temperature observed above the urban environment and key processes that affect nocturnal cooling."

The following text is added to be consistent with results shown in Fig. 12.

Line 861: "Above urban woods (about 900 ha) near-surface lapse rates can reach twice the value observed above urban parks (near 12°C/100m at 00 UTC). This leads to the development of an internal UBL, about 50 m (100 m) deep, above urban parks (woods)."

[Figure]

Figure 12. Typical night-time (near 00 UTC) vertical structure of wind speed (left) and potential temperature (right) in the urban boundary layer, observed during a) stagnant, b) intermediate, and c) turbulent mixing regimes. Potential temperature profiles are represented above built-up environments (red), urban parks (yellow) and urban woods (green). Key processes (advection, turbulent mixing, radiative cooling) affecting nocturnal cooling in the UBL are represented (centre).

RC1 – Comment 2:

2、 Suggestion: If possible, in the future, it is recommended that the authors collect observational data from other cities to analyze whether cities with different terrains and climates also reach similar conclusions. This would be a very meaningful study.

Author Reply to comment 2:

Indeed, it will be interesting to study the occurrence of turbulence regimes in other cities with different geographical settings and climate conditions. Our next study is to refine our results with more parks to study the impacts of park dimensions, locations, taking into account when possible soil moisture, vegetation types …

RC1 – Comments 3:

Author Reply to comment 3:

All suggested changes have been implemented in the manuscript.

3、line 25: "near-by" should be "nearby"
4、line 61: "Trees can provide efficient shading whereby reducing daytime air temperatures"
should be "Trees can provide efficient shading, reducing daytime air temperatures"
5、line 65: "The green infrastructures" should be "Green infrastructures"
6、line 79: "e.g" should be "e.g."
7、line 190: "Due to installation setup" should be "Due to the installation setup"
8、line 239: "i.e." should be "i.e.,
"
9、line 332-333: "the advection" should be "advection"
10、line 335: "July 19" should be "19 July"
11、line466: "three turbulence regimes" should be "three types of turbulence regimes"
12、line 610-611: "sensible heat fluxes" should be "sensible heat flux"
13、line 768: "the temperature at all sites decreases to reach a minimal value the next morning
at sunrise." Should be "the temperature at all sites decreases to a minimal value by the next morning at sunrise."
14、line 811: "is" should be "are"
15、line 848: "Hence in the" should be "Hence, in the"

**RC2 review**

The authors would like to acknowledge the time and effort spent by the reviewer RC2 on our manuscript. We provide replies to each reviewer comment below. All replies and revised text are shown in blue in this document.

RC2 – General comment:

In this work, the nocturnal cooling effect of urban parks within Paris is investigated using an excellent suite of measurements, which allows the authors to concurrently examine the meteorological conditions of the surface and lower atmosphere. Three stability regimes are identified, quantified, and their implications are summarized. The meteorological factors are put in context both spatially and temporally throughout the night. The manuscript has clear motivations and implications, is logically organized, well-written, and includes appropriate figures. There are only a few minor comments.

RC2 – Comment 1:

The lowest gate (~240 m AGL) is used for the vertical variance, but in Fig. 6e/7e it appears to be either at an inversion base or perhaps just above it in these examples. Is there potential that being just below or above the inversion base would have a significant impact on the analysis or conclusions? Although the limitations of the instrumentation prevents measurements closer to the surface, it might be good to at least comment on this point since any measurement near a sharp change of temperature gradient could have large differences just above or below.

Author Reply to comment 1:

We agree with the reviewer that the lowest gate at which we can retrieve vertical velocity variance is close (just below or just above) to the height of the temperature inversion, that also correspond to the height of peak horizontal wind speed.

In Fig. 5d/6d/7d/8d we only show the vertical velocity variance values at one height (240m AGL), but we have access to vertical profiles of vertical velocity variance from 240 m AGL up to 2000 m AGL or more. We observe that vertical velocity variance values vary significantly between 240m and about 500m AGL, in the layer where the low-level jet is observed, with peak values at heights where the wind speed is maximum. Under stagnant regime conditions, the vertical velocity variance values are significantly reduced in the 240-500 m AGL layer (i.e. values < 0.01 $m^2$ $s^{-2}$), compared to intermediary (i.e. values > 0.05 $m^2$ $s^{-2}$) or turbulent (i.e. values > 0.1 $m^2$ $s^{-2}$) regimes. Hence, we conclude that the vertical velocity variance value at 240 m AGL is representative of the nocturnal urban boundary layer turbulence regime.

We add the following sentence in the manuscript Line 528: "It should be noted that in the stagnant regime, the vertical velocity variance values are very low (less than 0.05 $m^2$ $s^{-2}$) throughout the LLJ layer that extends from 240 to about 500 m AGL or more (not shown). Hence even though vertical velocity variance is not constant with height and the

measurement height (240 m AGL) is close the top of the UBL, we conclude that the vertical velocity variance value at 240 m AGL is representative of the nocturnal urban boundary layer turbulence regime".

RC2 – Comment 2:

Related to the lowest gate, it is given as both 238 and 240 m AGL in the text, but please just select one to be consistent (either the 238 or the rounded 240).

Author Reply to Comment 2:

We harmonized the text with the lowest Doppler Lidar gate at 240 m AGL. 238 m was replaced with 240 m four times in the manuscript.

RC2 – Comment 3:

Soil moisture is mentioned around line 742, but given the impact it could have on heating/cooling processes associated with the park, it might be beneficial to comment on this earlier. Specifically, it should be stated if this period was during wet, normal, or drought conditions, and also if these parks are irrigated.

Author Reply to Comment 3:

Indeed precipitation and irrigation conditions will have an impact on plant health and their evapotranspiration capacity which contributes to about 30% of the daytime cooling effect of trees for example (70% is due to shading). Summer 2022 in Paris was both hotter and dryer than a normal year. Irrigation practices differ from one park to another and also inside each park, but quantified information is not available. As suggested, we clarify this in the paper as follows:

Line 238: On average this period is characterized by a positive temperature anomaly and near-zero precipitation anomaly (not shown).

Line 253: The three selected urban parks differ however in terms of vegetation type (species; fractions of trees, shrubs and grass) and also in terms of irrigation practices and hence soil moisture. These differences and their effects are not accounted for in this study.

**Additional unsolicited revisions:**

The following reference was missing in the reference list and was added line 983

Gao, Z., Zaitchik, B. F., Hou, Y., & Chen, W. (2022). Toward park design optimization to mitigate the urban heat Island: Assessment of the cooling effect in five US cities. Sustainable Cities and Society, 81, 103870. https://doi.org/10.1016/j.scs.2022.103870